# Fermentation Strategies for Production of Pharmaceutical Terpenoids in Engineered Yeast

**DOI:** 10.3390/ph14040295

**Published:** 2021-03-26

**Authors:** Erdem Carsanba, Manuela Pintado, Carla Oliveira

**Affiliations:** 1Amyris BioProducts Portugal, Unipessoal, Lda. Rua Diogo Botelho 1327, 4169-005 Porto, Portugal; carsanba@amyris.com; 2CBQF—Centro de Biotecnologia e Química Fina—Laboratório Associado, Universidade Católica Portuguesa, Escola Superior de Biotecnologia, Rua Diogo Botelho 1327, 4169-005 Porto, Portugal; mpintado@porto.ucp.pt

**Keywords:** terpenoids, *S. cerevisiae*, pharmaceutics, fermentation, fed-batch

## Abstract

Terpenoids, also known as isoprenoids, are a broad and diverse class of plant natural products with significant industrial and pharmaceutical importance. Many of these natural products have antitumor, anti-inflammatory, antibacterial, antiviral, and antimalarial effects, support transdermal absorption, prevent and treat cardiovascular diseases, and have hypoglycemic activities. Production of these compounds are generally carried out through extraction from their natural sources or chemical synthesis. However, these processes are generally unsustainable, produce low yield, and result in wasting of substantial resources, most of them limited. Microbial production of terpenoids provides a sustainable and environment-friendly alternative. In recent years, the yeast *Saccharomyces cerevisiae* has become a suitable cell factory for industrial terpenoid biosynthesis due to developments in omics studies (genomics, transcriptomics, metabolomics, proteomics), and mathematical modeling. Besides that, fermentation development has a significant importance on achieving high titer, yield, and productivity (TYP) of these compounds. Up to now, there have been many studies and reviews reporting metabolic strategies for terpene biosynthesis. However, fermentation strategies have not been yet comprehensively discussed in the literature. This review summarizes recent studies of recombinant production of pharmaceutically important terpenoids by engineered yeast, *S. cerevisiae*, with special focus on fermentation strategies to increase TYP in order to meet industrial demands to feed the pharmaceutical market. Factors affecting recombinant terpenoids production are reviewed (strain design and fermentation parameters) and types of fermentation process (batch, fed-batch, and continuous) are discussed.

## 1. Introduction 

Terpenoids, also known as isoprenoids or terpenes, constitute one of the largest chemically and structurally diverse group of natural products with over 80,000 members in a widespread family (currently stated in the Dictionary of Natural Products database, http://dnp.chemnetbase.com, accessed on 26 March 2021) [1,2,3,4]. Although the majority of terpenoids are predominantly present in plants, other organisms such as bacteria, fungi, insects, and animals contain them as well [5]. Their essential biological functions can be counted as the regulation of cell growth and defense in plants (e.g., tocopherol, gibberellin), and intracellular electron transport, glycoproteins biosynthesis and cell membranes formation in animals (e.g., cholesterol) [6]. Owing to their structural variety, there exist diverse terpenoid activities, such as mediating symbiotic or antagonistic interactions between organisms to electron transfer, protein prenylation, or contribution to membrane fluidity, which make these compounds highly useful to be applied in various industries: food, cosmetics, fine chemistry, pharmacy, agriculture, and biofuel [5].

Terpenoids are formed from several isoprene (2-methylbuta-1,3-diene; IUPAC 1997) (CH_2_=C(CH_3_)–CH=CH_2_) structural units (or building blocks) synthesized from mevalonate (MVA) or 2-C-methyl-d-erythritol 4-phosphate (MEP) (which is also known as the non-mevalonate) pathway [2,3]. Through the condensation reactions of these units, several large and more complex compounds are produced. According to the number of C5 isoprene units, terpenoids are mainly classified in hemiterpenoid (C5), monoterpenoid (C10), sesquiterpenoid (C15), diterpenoid (C20), triterpenoid (C30), tetraterpenoid (C40), and polyterpenoid (C > 40) [2,3,6]. Terpenoids are mostly formed through continuous addition of head to tail manner of building blocks, which are isoprene diphosphate (IPP) and dimethylallyl diphosphate (DMAPP). For instance, the precursor of monoterpenoids, geranyl diphosphate (GPP), is produced by head to tail condensation of these two building blocks. Later, addition of IPP leads to the formation of farnesyl diphosphate (FPP) and geranylgeranyl diphosphate (GGPP), which are precursors of sesquiterpenoid and diterpenoid. On the other hand, head to head condensation of two molecules of FPP and GGPP, respectively, forms squalene and phytoene, the precursors of triterpenoid and tetraterpenoid [6,7] (Figure 1). Furthermore, these precursors are exposed to oxidation, by cytochrome P450 oxygenases, and glycosylation, by glycosyltransferases to form various terpenoids. In addition, prenyltransferases concatenates isoprene-derived precursors to fatty acid-derived precursor to synthase complex terpenoids or meroterpenoids, which consist of the medically important compounds, cannabinoids [2].

From early ancient times up to now, terpenoids have been playing important roles in many medical treatments as pharmaceuticals with diverse biomedical activities, such as antimalarial, anticancer, anti-inflammatory, antibacterial, antiviral, hypoglycemic, preventing and treating cardiovascular diseases, and promoting the transdermal absorption. Besides that, they can have insecticidal, immunomodulatory, antioxidant, antiaging, and neuroprotective effects [2,3,8,9]. They are usually produced from natural resources through extraction or by chemical synthesis, but the obtained yields are generally low due to inherently low level of target compound and the necessity of complex extraction and chemical synthesis methods, which are commercially infeasible [10]. In most cases, the natural production method, extracting terpenoids from original sources (e.g., taxol from yew tree and artemisinin from the plant *Artemisia annua*) generally fails in terms of quality and supply management due to seasonal and geographical alterations [5]. In addition, plant engineering for terpenoid production is difficult and complex due to tissue specific expression and loss of volatile products by evaporation, and productivity and yields are very low [11]. Due to these limitations, microbial production of terpenoids has received increasing attention, since production of these compounds at large scale fermentation by engineered microorganisms offers a promising higher yield, batch-to-batch consistence, lower production cost, and more sustainability.

Among terpenoids, the sesquieterpenoid artemisinin have been often used as an antimalarial drug and the diterpenoid taxol (paclitaxel) have been developed to be an important anticancer chemotherapy drug for many years [12]. Semi-synthetic artemisinin is currently manufactured by the French pharmaceutical company Sanofi, using engineered *Saccharomyces cerevisiae* strain developed by Amyris [13], which is a very important example of microbial industrial production of terpenoids. However, the same success has not been yet achieved for paclitaxel due to the complexity of its synthesis pathway, which is still unclear and further studies are required to fully elucidate it [14]. So far, the highest recorded titer of oxygenated taxanes has reached up to 570 mg/L in engineered *Escherichia coli* by optimizing the P450 expression of taxanes and other related enzymes [15]. 

For centuries, the baker’s yeast, *S. cerevisiae*, has been mainly used in the industrial production of alcoholic beverages (wine, beer, and distilled spirits), bakery products, and bioethanol. However, with the latest developments in synthetic biology, it became one of the most widely industrially used cell factory in the microbial production of a wide variety of products, such as alcohols, organic acids, amino acids, enzymes, therapeutic proteins, chemicals, and metabolites [16]. Among them, for example, biopharmaceutical recombinant peptide hormone, insulin, has been produced by genetically engineered *S. cerevisiae* strains for many years. Many pharmaceutical companies have chosen this yeast as the most suited host organism to produce a large variety of recombinant products due to its well-known genetics, physiology, biochemistry, and genetic engineering background, the availability of genetic tools, and the suitability of dense and large scale fermentation [16,17,18]. 

In the same line, *S. cerevisiae* has emerged as a model organism for the production of terpenoids since it has many additional advantages other than mentioned above, such as generally regarded as safe (GRAS) status, high genetic tractability, ease of manipulation, possessing universal endogenous MVA pathway, ability to express eukaryotic cytochrome P450 enzymes, robustness, relatively absence of secondary metabolites, high sugar catabolic, fast growth rate, and high tolerance against harsh industrial conditions [7,19,20,21,22].

Besides *S. cerevisiae*, other microorganisms have been explored for terpenoids production. Among them, *E. coli* has the most restrict chassis, since its produces natively limited amounts of terpenoids (e.g., quinones) and, therefore, the improvement of MEP pathway by engineering enzymes for IPP and DMAPP synthesis, or the introduction of heterologous MVA pathway, is required [23]. In contrast, *S. cerevisiae* has an endogenous MVA pathway, producing high amounts of ergosterol and native cytochrome P450 enzymes for the modification of terpenoids skeleton. Nonconventional yeast *Yarrowia lipolytica* has been also considered as a suitable yeast to synthesize terpenoids due to its capacity to produce large amount of acetyl-CoA, the initial substrate of the MVA pathway [23]. In addition, carotenogenic yeast *Rhodosporidium toruloides* can naturally accumulate several carotenoids (C40 terpenoids), indicating that it might have high carbon flux through MVA pathway, ensuring pools of intermediates for producing diverse types of terpenes [24]. This yeast can metabolize efficiently both xylose and glucose, and tolerates high osmotic stress, enabling the use of lignocellulosic hydrolysates as feedstock in contrast to *S. cerevisiae* [24]. Cyanobacteria have also the potential to produce sustainable terpenoids using light and CO_2_ instead of sugar feedstocks. However, terpenoids titer and productivity obtained are still below industrial levels and further studies to overcome the barriers for efficient conversion of CO_2_ to terpenoids are needed [25]. 

Overall, *S. cerevisiae* has as main advantage over *E. coli* and cyanobacteria hosts its intrinsic MVA pathway, and the disadvantage over *Rhodosporidium toruloides* host the incapacity of using directly lignocellulosic hydrolysates as feedstock. Nevertheless, *S. cerevisiae* is quite superior to the other microorganisms in respect to higher process robustness, fermentation capacity, plenty of available genetic tools in pathway engineering and genome editing, and proven capacity to attain industrial levels of relevant terpenoids [23]. 

To date, there has been a strong effort for terpenoid biosynthesis through metabolic engineering of microbes, however, production levels are at the mg/L scale in scientific literature, which are generally too low and commercially insufficient. Economically meaningful metrics of titer (g product per L broth), yield (g product per g substrate), and productivity (g product per L broth per hour) should be provided for industrial production [11]. Fermentation development at scale has a crucial importance to improve terpenoids production. For example, Amyris has reached titers of more than 130 g/L of β-farnesene and 25 g/L of artemisinic acid (precursor of artemisinin, antimalarial drug) from sugar cane feedstock in engineered yeast *S. cerevisiae* through optimized fed-batch fermentation [26,27,28]. Fermentation strategies can increase productivity and reduce the cost of production via improving medium composition, optimizing physicochemical conditions, and applying efficient downstream processing. However, a complete overview of the current approaches for obtaining terpenes relevant for the field of pharmaceuticals by yeast fermentation has not yet been reviewed in the literature. Therefore, this review details the production of pharmaceutical terpenoids by engineered yeast *S. cerevisiae* and focuses attention on fermentation strategies to improve their production scale. Different fermentation factors and processes are discussed. 

## 2. Pharmaceutical Terpenoids

A vast number of terpenoids have been widely used in medicine and medical sciences to prevent and treat many diseases due to their pharmaceutically bioactive properties. Their wide pharmaceutic effects and medical functions have already been extensively revised in the literature (see reviews: [3,7,8,9,14,29,30,31]. Some of the pharmaceutical terpenoids currently used in medicine and produced via biotechnological approaches are presented in Table 1.

### 2.1. Artemisinin and Its Derivatives

The sesquiterpene endoperoxide artemisinin, which is isolated from wormwood (*Artemisia annua*), is the most effective traditional antimalarial drug with the property of high efficiency and low toxicity. It shows to be effective against a broader range of life cycle stages of the apicomplexan parasite than conventional antimalarials such as quinine. Antimalarial property of artemisinin arises from its extraordinary endoperoxide ring, which has an impact on overcoming *Plasmodium* spp. infections (Figure 2) [3,8,39,40]. The mechanism of action is still unclear, but there are some possible explanations. Right after the infection by *Plasmodium* spp., phagocytosis of red blood cells occurs and heme proteins in high levels are delivered. High concentration of heme stimulates the artemisinin, which later binds to the parasite proteins in the *Plasmodium* spp. body to inactivate them and consequently eradicate the infection of *Plasmodium* spp. [39,41]. 

Besides antimalarial activity, artemisinin and its derivatives have anticancer activity in vivo and in vitro, with small adverse effects. For instance, it has been reported that artesunate, which is a semisynthetic derivative of artemisinin, displays antitumor activity against several cancers, such as breast, lung, prostate, colon, ovarian, melanoma, and leukemia cancers [42]. Moreover, antibacterial and antiviral activities of these compounds were also reported in several works. Artemisinin and its derivatives have concentration dependent antibacterial activities against anaerobic, aerobic, facultative anaerobic and microaerophilic bacteria [43,44]. They also exhibited significant antiviral activity against the hepatitis B (HBV) and hepatitis C viruses [45]. Finally, artemisinin has been reported to have hypoglycemic effect and to be able to improve type 1 diabetes [8].

### 2.2. Paclitaxel (Taxol)

Paclitaxel, also known as its commercial brand name “Taxol”, is a tricyclic diterpenoid compound (Figure 3) used as chemotherapy drug. It was first isolated from bark and needle of the pacific yew tree (*Taxus brevifolia*) and has effective anticancer activities on several type of tumors, such as ovarian and breast cancers. Its unique mechanism of action as a microtubule stabilizer and inhibitor of mitosis on ovarian and breast cancers was approved by the US Food and Drug Administration (FDA) in 1992 and 1994, respectively. Since then, Paclitaxel has been also widely used in the treatment of colorectal, head and neck, small-cell and non-small-cell lung cancers (NSCLCs), and acquired immune deficiency syndrome (AIDS). It contributes the conjunction of tubulin into microtubules and hinders the breakage of microtubules, which inhibits cell cycle progression, stops mitosis, and prevents the generation of tumor cells [3,8,14,33].

Paclitaxel is very expensive due to its small concentration found in original plant source (yew tree). For instance, 10,000 kg bark of yew trees are required to produce 1 kg paclitaxel. This amount can treat 500 patients but requires the harvest of 300 yew trees [14,46]. Production is dependent on slow growth and threat of species and there is the risk of yew trees extinction. Due to these reasons, alternative sources are necessary to be developed. Until now, several other options, such as microbial production, semi-synthesis and artificial cultivation of *Taxus brevifolia* were applied to meet world demand of Paclitaxel [8,14]. Moreover, the production and extraction of this compound from genetically engineered endophytic fungi has been recently announced to be an effective way as well [46]. 

### 2.3. Cannabinoids

Cannabinoids comprise a group of more than 100 prenylated phenolic compounds found in *Cannabis* spp. (Cannabaceae), mainly in the plant *Cannabis sativa*. Apart from plant derivatives, Cannabinoids also involve Endocannabinoids, Phytocannabinoids, and Synthetic cannabinoids, which are able to bind to the human cannabinoid receptors [8,34,47]. Cannabinoid rich plants like *C. sativa* and *C. indica* have been used for different purposes for more than 5000 years. The fiber-type of Cannabis (hemp), which contains the major cannabinoid, cannabidiol (CBD) (Figure 4), has been often used in the treatment of pain relieving conditions. On the other side, Δ9-tetrahydrocannabinol (THC) (Figure 4), found in high quantity in Cannabis (marijuana), has been known for psychoactive properties, and generally used as a recreational drug and in the treatment of several medical conditions [8,48,49]. In addition, many pharmaceutical companies have produced several synthetic cannabinoid drugs, known by different brand names, such as Cesamet (Valeant Pharmaceuticals North America, Bridgewater, NJ, USA), Marinol (Unimed Pharmaceuticals Inc., Marietta, GA, USA), and Sativex (GW Pharmaceuticals plc, Cambridge, UK). These drugs have been used in Canada, USA, UK, and other countries to treat nausea and vomiting caused by cancer chemotherapy, anorexia associated with weight loss in patients with AIDS, and symptomatic relief of neuropathic pain in multiple sclerosis [34,50].

Cannabinoids also called as terpenophenolic in the class of meroterpene, are produced as secondary metabolite of Cannabis through concatenation of isoprenoid precursor with a second fatty acid derivative precursor by a prenyltransferase enzyme. This enzyme synthesizes cannabigerolic acid (CBGA), through the prenylation of olivetolic acid and geranyl diphosphate (GPP). CBGA is then converted into different cyclized compounds, such as tetrahydrocannabinolic acid (THCA), cannabidiolic acid (CBDA), and cannabichromenic acid (CBCA). Finally, these compounds undergo decarboxylation to produce biologically active compounds THC, CBD, and CBC, which are the main cannabinoids placed in *Cannabis* spp. [2,8,34,51].

In recent years, *Cannabis* spp. has been legally used for the medical treatments in many countries, such as USA, Canada, Israel, and several European countries including the Netherlands, Germany, and the Czech Republic. However, *Cannabis* spp. cannot be cultivated in several countries due to the legal issues. Agricultural production of cannabinoids via Cannabis cultivation has several disadvantages, such as low yield due to climate changes and plant diseases, low concentrations of cannabinoids in plant, necessity of extraction processes, and sociopolitical factors for the use of Cannabis, which has been mostly seen as a source of narcotics. On the other hand, chemical synthesis of cannabinoids does not offer an alternative production way because of synthesis complexity, which generally causes low yields and high production cost. For these reasons, microbial production of these compounds by engineered yeast would serve a sustainable, reliable, eco-friendly, and cost-effective alternative approach [2,34]. 

### 2.4. Other Medically Important Terpenoids

Apart from artemisinin, taxol, and cannabinoids, there are also other medically important terpenoids with several pharmaceutical activities and functions. It was reported that the monoterpenes perillyl alcohol, D-limonene, and geraniol are effective in the prevention and treatment of various types of cancers (Figure 5) [3,52,53]. They have therapeutic or prophylactic effects on breast, lung, colon, prostate, pancreatic, and liver cancers [3]. Their main mechanism to prevent cancer activity consists in the inhibition of posttranslational isoprenylation of proteins, which are responsible for the growth of tumor cells [52]. Besides antitumor activity, some monoterpenes (e.g., menthol, limonene, and sabinene) have antimicrobial effects on diverse type of microorganisms, such as *Bacillus subtilis*, *Staphylococcus aureus*, *Streptococcus* spp., *E. coli*, and *Candida albicans* [3,54]. In addition, menthol and limonene promote the transdermal absorption of drugs through the human skin [3,54]. These terpenoids show little skin irritation, low toxicity, and high activity. The sesquiterpene patchouli alcohol also presented antibacterial activity on *Helicobacter pylori* [32]. In addition, triterpene ginsenosides have various pharmacological effects, such as anti-oxidation, anti-inflammatory, hepatoprotection, anti-diabetic (hypoglycemic activity), and antitumor [36]. They also exhibit therapeutic effects on the prevention and treatment of various cardiovascular diseases, such as regulation of vascular function, inhibition of cardiomyocyte hypertrophy and thrombosis [3]. In addition, betulinic acid (Figure 5) and its semisynthetic derivatives, such as PA-457, have also shown remarkable pharmacological properties, including inhibitory effects against human immunodeficiency virus (HIV) and cytotoxicity activity on several type of cancer cells (Table 1) [37,38]. Very interestingly, biotransformation of betulinic acid has been continuously investigated aiming at discovering novel derivatives for pharmacological studies [55].

## 3. Biosynthesis of Medically Important Terpenoids

Terpenoid biosynthesis can be divided into four main stages. In the first stage, universal building blocks isopentenyl diphosphate (IPP) and dimethylallyl diphosphate (DMAPP) are synthesized via the mevalonate pathway (MVA pathway) and the methyl-erythritol phosphate pathway (MEP pathway). MVA pathway, which exists in *S. cerevisiae*, starts from acetyl-CoA and ends up with the production of DMAPP. In the second stage, geranyl pyrophosphate (GPP), farnesyl pyrophosphate (FPP), or geranylgeranyl pyrophosphate (GGPP), which are the precursors of monoterpenoids, sesquiterpenoids, or diterpenoids, respectively, are created by condensation reactions of one molecule DMAPP with one, two, or three molecules of IPP under the action of isopentenyl transferase. On the other hand, triterpenoids are similarly generated via polymerization of two molecules of FPP. In the third stage, these linear isopentenyl pyrophosphate precursors are cyclized or rearranged by terpene synthases to form primary terpene carbon skeleton. Finally, this carbon skeleton undergoes several post-modifications and tailoring reactions (oxidations, acetylation, esterification, alkylation, etc.) under the action of cytochrome P450s (CYPs) to form biologically active terpenoids (for recent reviews see: [5,12,56]).

## 4. Metabolic Engineering Strategies for Pharmaceutic Terpenoids Production in Yeast

A brief overview of metabolic engineering applications in *S. cerevisiae* for terpenoid biosynthesis is herein given. The common metabolic engineering strategies for terpenoids synthesis in yeast, as described in recent reviews [5,7,56,57,58] can be summarized as such:Modifying endogenous pathways for synthesis of desired terpenoids.Finding and introducing new heterologous enzymes and pathways into yeast.Determination of rate limiting steps in selected pathways by application of omics studies.Elimination of rate limiting steps in target pathways via overexpression of genes, and cofactor and transporter engineering.Developing enzyme activity and/or specificity by protein engineering.Improving expression level of target key enzymes.Blocking or down regulating competing pathways.Increasing precursor and cofactor supply.Balancing cell growth and terpenoid synthesis for fermentation process.

As above mentioned, the direct precursors in sterol pathway, FPP, GPP, GGPP, and squalene epoxide, can be converted into specific terpenes through a series of enzymes designated as terpene synthases, which do not exist in *S. cerevisiae*. For this reason, heterologous genes for the functional expression of terpene synthases from plant resources (e.g., taxadiene synthases from *Taxus brevifolia*) or synthetic sources (from NCBI gene accession number) should be introduced into *S. cerevisiae* [57]. On the other hand, Cytochrome P450 (CYPs), which are essential enzymes to modify the hydrocarbon products from terpene synthases, catalyze a wide variety of metabolic reactions, such as oxygenations, deamination, decarboxylation, dealkylation, and C–C cleavage [59]. *S. cerevisiae* has only three CYPs, which are Erg5 and Erg11 in the ergosterol pathway and Dit2 (putative cytochrome P450 involved in the synthesis of *N*,*N*-bisformyl dityrosine). For specific or non-native terpenoid production, heterologous expression of CYPs is required. As a successful example, ref. [27] have expressed CYP71AV1, NADPH dependent cytochrome P450 oxidoreductases (CPR1) and cytochrome b5 (CYB5) from *A. annua* in *S. cerevisiae*, obtaining 25 g/L of artemisinic acid in optimized fed-batch fermentation.

Another strategy is to identify the rate limiting reactions and overexpress the gene, or genes, responsible for these reactions to promote flux into the MVA pathway of *S. cerevisiae*. HMG-CoA reduction to MVA, catalyzed by Hmg1p or Hmg2p, products of HMG1 and HMG2, respectively, was reported as the rate limiting step in the sterol pathway of *S. cerevisiae*. Overexpression of N-terminal truncated HMG1 (tHMG1) and Hmg2p in anaerobic conditions increased squalene titer in this yeast [7,57]. Moreover, overexpression of the transcriptional factor Upc2-1, constitutively active mutant of Upc2, improved sterols uptake [7]. As a similar strategy, overexpression of all structural genes in MVA pathway (ERG10, ERG13, tHMG1, ERG12, ERG8, IDI1, and ERG20) has enhanced terpenoid production [26,60]. In addition, down-regulation of downstream genes of competitive pathways can increase the precursor pool. For instance, down regulation of ERG9 in low glucose concentrations through HTX1 promoter, could increase FPP pool in fed-batch fermentations [61].

Acetyl-CoA and NADPH are the essential precursor and cofactor for the biosynthesis of several products (sterols, fatty acids, and polyketides) as well as terpenoids. Overexpression of structural genes in MVA pathway in *S. cerevisiae* to improve terpenoid synthesis results in draining the acetyl-CoA and NADPH pools, which may negatively affect the cell growth, causing in parallel the decrease in the terpenoid production. Therefore, the increase of cytosolic acetyl-CoA and NADPH pools is required. The most effective way for achieving this was to establish a synthetic pathway. Ref. [26] combined acetaldehyde dehydrogenase (which converts acetaldehyde to acetyl-CoA) with xylulose-5-phosphate-specific phosphoketolase and phosphotransacetylase (which convert xylulose-5-phosphate to acetyl-CoA) to increase acetyl-CoA pool. Moreover, the same authors have used an NADH-consuming HMG-CoA reductase to increase NADPH pool. This new pathway lead to improved β-farnesene production along with 75% less oxygen requirement in fed-batch fermentation [26].

Interestingly, to expand the diversity of terpenoid structures, C11 terpene scaffolds were produced in *S. cerevisiae* by engineering dedicated synthases upon identification of a single residue switch that converts C10 plant monoterpene synthases to C11 specific enzymes [62]. More recently, a synthetic orthogonal monoterpenoid pathway based on an alternative precursor, neryl diphosphate, was established in yeast, in which five engineered enzymes were combined with dynamic regulation of metabolic flux to take advantage of the orthogonal substrate potential and improve monoterpenes production [63].

## 5. Factors Affecting Fermentation Process of Pharmaceutical Terpenoids

The main factors affecting the production of terpenoids in *S. cerevisiae* include strain engineering, inoculum size, pH value, temperature, oxygen rate, and fermentation medium composition such as carbon and nitrogen sources. In addition, secondary products like ethanol, and some organic acids can affect the productivity of terpenoids. Below, the general factors affecting terpenoid biosynthesis are discussed.

### 5.1. Strain Engineering

Wild type *S. cerevisiae* strains (most known laboratory strains used for system biology: CENPK, S288C, BY4741 and W303, www.yeastgenome.org, accessed on 26 March 2021) have endogenous sterol biosynthesis process, including MVA pathway. Ergosterol is synthesized through this process. By using molecular biology methods, namely gene deletion and high-through screening methods, all enzymes and genes upstream the MVA pathway, and downstream ergosterol synthesis in *S. cerevisiae*, have been clarified [64]. Terpenoids cannot be directly produced by wild type strains of *S. cerevisiae* without genetic modifications because it requires the availability of precursors for transfer of intermediates between compartments (the cytoplasm and endoplasmic reticulum) and diverse class of Cytochrome P450 (CYP-CPR) enzymes for the oxygenation reactions to produce structurally diverse terpenes [57]. In order to produce terpenoids efficiently in yeast, several genetic modifications such as promoter alterations, gene mutations, genes knockout and expression of heterologous genes were required (see recent reviews: [2,6,12,65]. Highly engineered strains of *S. cerevisiae*, with the capability to produce different type of terpenoids, including industrial products, such as farnesene, artemisinin, patchouli alcohol, squalene, geraniol, and β-carotene, have been constructed [5]. However, different gene regulation is necessary for different type of end-products. For instance, in sesquiterpenes production, ERG9 enzyme is required to be downregulated to lower the metabolic flow from farnesyl pyrophosphate to squalene, while in triterpenoids production, the same enzyme is upregulated to increase precursors pull for triterpenoids [64]. In addition, distinctive plant derived cytochrome P450 enzymes for specific target products should be cloned and expressed as well [12]. Because of these reasons, tailor-made engineered strains need to be constructed for specify target terpenoids [7,56]. Some of the important genetic modifications for different terpenoid biosynthesis are presented in Table 2.

### 5.2. Carbon Source

The selection of fermentation carbon source depends on the precursor pool designed to be used [60]. Ethanol was often fed to fermentation broth as substrate since it can be readily directed to acetyl-CoA formation [69]. It was preferred instead of glucose as carbon source during the production phase due to increased higher titer and yield in the accumulation of many terpenoids in *S. cerevisiae*, such as amorpha-4,11-diene (precursor of artemisinin) [60], artemisinic acid [27], β-amyrin [71], limonene [72], geraniol [73], protopanoxadiol [74], and patchoulol [22] (Table 3). In these studies, diauxic yeast fermentation process was performed, in which glucose is consumed in the first stage of cell growth with the simultaneous production of ethanol that is then used in the second stage of cell growth and biosynthesis of the product of interest [72]. However, high ethanol concentration results in cell stress in *S. cerevisiae.* As a response to this situation, ergosterol synthesis is usually stimulated. This metabolic mechanism had improved the availability of the main precursor 2,3-oxidosqualene, resulting in two times higher total triterpenoid productivities and 2.4-fold increase in carbon yield in ethanol pulse fed fermentations, compared to glucose fed fermentations [75]. However, the use of ethanol as main carbon source is not feasible for industrial terpenoid production, due to its high cost as compared to other carbon sources like glucose. To date, glucose, which is the most widely used carbon source in bioprocesses, has been also selected as feedstock to reach high titer and yield during fermentation of several terpenoids, such as bisabolone [76], zerumbone [77], α-humulene [4], α-santalene [61], miltiradiene [78], protopanaxadiol and dammarenediol-II [79], and ginsenoside Rh2 [35] (Table 3). In addition, the promoters chosen to drive expression of heterologous genes also affect the choice of the substrate. For instance, when gal-promoters were used in the construction of engineered yeast, galactose was required to be included in the medium in order to induce the expression of cloned genes, while being used as carbon source at the same time [57]. Some terpenoids were produced using galactose as sole carbon source, such as limonene [80], (-)-limonene [81], sabinene [70], bisabolene [66], valerenic acid [82], and polpulonic acid [20] (Table 3). However, using galactose as sole carbon source makes the process expensive and regulation by galactose can be only repressed in the presence of glucose. Because of this, the Gal1 gene was deleted to produce artemisinic acid from glucose as a primary carbon source instead of galactose, where galactose was supplemented in low amount as a gratuitous inducer [60]. Apart from these carbon sources, molasses, which consists mainly of sucrose, disaccharide composed of glucose, and fructose, is commonly used as carbon source in many industrial fermentations [17,83]. Ref. [84] produced nerolidol from sucrose as sole carbon source in carbon-overflow fed-batch fermentation by employing the diauxic-induction system, including the four characterized GAL promoters. Moreover, besides these carbon sources, raffinose and dextrose were also used in a mixture with galactose or glucose in the production of polpulonic acid [20] and β-amyrin [85], respectively. However, from all employed carbon sources, the highest specific production rates were achieved when using ethanol in the diauxic yeast fermentation process [27,60,69]. Rather than these hexoses, disaccharide sucrose, as well as ethanol, agricultural byproducts like sugar cane bagasse or straw, which comprise mostly of xylose and glucose, as well as other sugars like galactose, can be considered as alternative feedstock for terpenoid bioprocess [7]. In addition, as a byproduct of biodiesel production, crude glycerol also has potential to be used as economic alternative source [86]. However, *S. cerevisiae* strains are unable to metabolize xylose and, thus, metabolic engineering must be performed, by introducing appropriate heterologous genes of xylose pathway in this yeast, to allow fermentation of this pentose. Significant advances have been made in recent years in regard of xylose use by yeast to obtain value-added bioproducts [87]. Overall, non-fermentable carbon sources (e.g., ethanol and glycerol) are more effective for attaining high terpenoid yields than fermentable ones (e.g., glucose, fructose, sucrose, and galactose) [86]. This is because the metabolic pathways of the non-fermentable carbon sources are highly flexible and involve the glyoxylate cycle and an increased number of mitochondrial shuttles in yeast [86].

### 5.3. Nitrogen Source

*S. cerevisiae* is able to utilize different kinds of nitrogen containing compounds in their natural habitat, such as amino acids, ammonium ions (NH_4_^+^), peptides, and urea. When these compounds are present in the growth medium, they are transported into the cells by permeases and used as building blocks in biochemical pathways or catabolized into glutamate and ammonium for nitrogen metabolism [91]. The type and concentration of nitrogen sources affect the growth and gene expression in this yeast. *S. cerevisiae* uses preferentially good nitrogen sources, such as ammonia, glutamine, and asparagine, over poor ones, such as proline and urea. Growth rate of this yeast is relatively high in good nitrogen sources rather than in poor nitrogen sources [92]. The accumulation of yeast biomass strongly depends on the nitrogen content in the growth medium, since it was reported that the proportion of nitrogenous compounds in yeast cells is about 50% (by weight) [93]. The most used nitrogen sources for terpenoid production are ammonium sulfate ((NH_4_)_2_SO_4_), ammonium dihydrogen phosphate ((NH_4_)H_2_PO_4_), ammonium chloride (NH_4_Cl), yeast extract, peptone, tryptone, and sometimes combination of those (Table 3). Higher terpene concentrations were achieved when (NH_4_)_2_SO_4_ was used as nitrogen source (Table 3). On the other hand, nitrogen limited resting cell fermentation enhanced the titer of betulunic acid (from 57 to 182 mg/L) and total triterpenoid concentration (from 319 to 854 mg/L) [75]. In this study, the NH_4_Cl concentration was decreased from 2.8 g/L to 0.939 g/L to allow nitrogen starvation. It was considered that restriction of biomass synthesis by nitrogen limitation might support the flux into terpenoid accumulation, since terpenoid synthesis has to compete for carbon, energy, and redox cofactors throughout yeast growth [94]. However, under this condition, cessation of growth and product formation, and lower specific productivity were observed in extended fermentations [75]. This indicates decrease in cell viability due to accumulation of toxic intermediates, or reactive oxygen species, which was also reported in the production of farnesene and artemisinic acid [27,95]. Deficiency of nitrogen content in prolonged fermentations can also result in sluggish or stuck fermentation. However, high nitrogen concentrations also carry risks to the process, once it may overstimulate yeast reproduction and increase biomass levels too fast, resulting in shortage of other yeast nutrients, increased fermentation temperatures and, eventually, causing stuck fermentations and productivity losses. Thus, prolonging the release of nitrogen sources under tight control during the fermentation process can be a good strategy to extend the fermentation period, without heat peaks and productivity breaks [96,97]. 

### 5.4. pH

In general, as an acidophilic organism, *S. cerevisiae* grows better under acidic conditions, in the pH range from 4 to 6, depending on the temperature, the presence of oxygen, and the strain [98]. The pH range between 5 and 6 has been mainly employed for terpenoid production (Table 3). In the study by [75], it was shown that fermentation time was reduced in batch cultivation at pH 6, while total terpenoid productivity decreased. Moreover, precipitation of solid particles, containing insoluble crystals and good quantity of triterpenoids on the inner wall of bioreactor was observed. The solubility of many terpenoids, especially hydrophobic triterpenoids, is pH-dependent, being higher at alkaline pH values, decreasing at lower pH values [99]. In another study, alkaline medium with pH ranging from 7 to 8 enhanced farnesol accumulation by *S. cerevisiae* [100]. It was also reported that the optimum pH for the wild type *S. cerevisiae*, and mutant strain in isopentenyl diphosphate isomerase activity (one of the important rate-limiting enzymes in terpenoid production), was 7.5 [101]. Alkaline pH might enhance the synthesis and solubility of terpenoids in the cultivation medium, but growth and biomass accumulation might be low, not allowing high cell density cultivation as desirable in industrial terpenoids production.

### 5.5. Temperature

The most used incubation temperature in terpenoid production is 30 °C. Rarely, 28 °C and 33 °C had also been employed, such as in the synthesis of geraniol and geranylgeraniol by *S. cerevisiae*, respectively (Table 3). In a study of the effects of fermentation temperature on *S. cerevisiae*, the highest population was achieved at 30 °C, while at lower temperatures, it was reached later [102]. In contrast, at high temperatures, especially at 35 °C, cell viability decreased [102]. High temperatures cause yeast stress, and, thus, such a condition is not favorable for its growth. As a response to the high temperature, heat-shock proteins are synthesized, and ribosome is inactivated. Different enzymes responsible for many microbial activities are also sensitive to high temperatures and are usually inactivated in this harsh condition [103]. Moreover, the role of temperature in (+)-valencene accumulation was investigated and found that remarkable enhancement in (+)-valencene concentration at 25 °C (87.2 mg/L) was observed compared to at 30 °C (20.6 mg/L) [21]. The benefit of lowering the temperature in the production of other relevant pharmaceuticals in yeast has also been previously reported in the case of plant lectins expression in *Pichia pastoris* [104,105].

### 5.6. Dissolved Oxygen

Under aerobic conditions, yeast converts glucose to carbon dioxide and water to produce energy. However, at high glucose concentrations, it also produces ethanol from glucose, which is known as the “Crabtree effect”. In yeast cultivation, the biomass increases very fast in the exponential phase, accompanying with high amount of oxygen consumption, so dissolved oxygen (DO) decreases quickly. Simultaneously, ethanol is produced because of the Crabtree effect. When glucose concentration decreases to a relatively low level, growth of yeast cells slows down due to inadequate carbon source in the medium; therefore, the DO increases promptly. If the glucose is added to culture medium, DO will again slowly decrease with the consumption of glucose. Thus, this approach of controlling DO plays an important role on cell growth and product formation, especially in fed-batch fermentations. The optimum glucose consumption rate and the yeast growth rate can be tightly controlled by the DO [106]. In a study for ergosterol production by fed-batch fermentation of *S. cerevisiae*, DO was a significant factor in its production and highest yield was obtained when DO was kept at 12%, explaining that the accumulation of this sterol was oxygen dependent [106]. Although oxygen is a vital component for aerobic organisms, it can also be a toxic agent that can damage cells by the action of reactive oxygen species (ROS). ROS can be generated through many different stress conditions, including high oxygen content in the culture medium [107]. Because of this, selection of the optimum DO level during terpenoid fermentation has significant effect on health and growth of cells as well as product formation. In many studies of terpenoid production, DO levels were kept above 30%, or 40%, by controlling agitation and aeration (cascade mode) in fed-batch processes (Table 3). During stationary phase of cultivation, DO levels reached up to 1.5%, which maintain micro aerobic fermentation in order to drive metabolic flux towards the production of fermentative products, like terpenoids [108].

### 5.7. Inoculum Size

The size of inoculum used for terpenoid production is generally applied between 2.5 % and 14% of medium volume or initial optical density (OD_600_) of 0.05 to 0.5 is used instead (Table 3). For instance, in a study for optimization of β-amyrin formation by engineered yeast in fed-batch fermentation, the most favorable inoculum size was reported as initial OD_600_ of 0.3 [109]. Moreover, in another study, inoculum size of 5.0 and 5.2% (*v*/*v*) with *S. cerevisiae* strain BY4741 and 8.1 and 2.6% (*v*/*v*) for strain EGY48 were favorable for squalene production in semi-anaerobic shake flask fermentations [110]. According to these results, optimum inoculum size was strain dependent. The difference between the productivity of these two strains at high inoculum size might be related with the synthesis of higher ergosterol by strain EGY48, which enhances tolerance to ethanol, which is produced during the early stages of fermentation, and which could be used as a substrate for squalene production under glucose depletion at the late stage of fermentation [110]. A reasonable high volume of inoculum is required to minimize the length of lag phase, increase specific growth rate, and accumulate the maximum biomass in the production fermenter in a shorter time as possible, which helps to achieve highest productivity. A study for inoculum size effect on metabolic regulation and stress response of *S. cerevisiae* in high cell density fermentation showed that as inoculum size (40 g/L initial biomass) increased, stress protectants (glycerol and proline) in glycerol biosynthesis and amino acid metabolism improved. However, citric acid cycle (TCA) intermediates are depressed and metabolites (myo-inositol and ethanolamine) associated with membrane structure and function decreased [111]. In the same study, growth rate, glucose consumption rate and ethanol productivity increased in high inoculum size, which is convenient for industrial ethanol fermentation. However, very high inoculum size will cause the rapid depletion of nutrients and oxygen, which results in microaerobic condition and increased oxygen demand. High ethanol production in anaerobic condition might have toxic effects on yeast cells and be the reason for the decrease in terpenoid yield. Low inoculum size causes longer fermentation phases, lower productivity parameters, and high risk of contamination. However, optimum inoculum size can help the culture to grow at extreme conditions, such as high salt concentrations and pH ranges [112,113].

## 6. Modes of Fermentation Process

Batch, fed-batch, and continuous fermentation process have all been used in terpenoids production. In the batch type of operation, required nutrients, feedstock, other ingredients, and microorganism are added to fermentation tank at the beginning of fermentation, while in the fed-batch type, some nutrients and ingredients are added to tank at certain intervals of time. In the continuous type of operation, input materials (nutrients and ingredients) are continuously fed and output materials (culture broth) are simultaneously removed from fermentation tank [83,103]. The choice of the most effective operation mode for terpenoids production depends on the fermentative properties of microorganism (e.g., kinetics, toxicity tolerance, lifespan), as well as the feedstock nature (e.g., sterilization, preparation, concentrations). 

Although metabolic engineering strategies, and pathways improvement, are employed to enhance terpenoids expression, high cell density cultivation is needed to obtain high product titers, yield, and productivity for industrial processes. For high cell density cultivation, it is also important to choose an optimum method of reactor operation, since environmental conditions affect the growth and product formation. Among these conditions, nutrient and substrate concentrations are important to be kept in specific ranges to avoid overfeeding or underfeeding [114]. 

### 6.1. Batch Mode

Batch operation mode is the simplest and easiest process since it requires low cost, low labor work, less control, as well as easy sterilization and preparation of feedstock [103]. Thus, it is usually preferred due to the practicability of recovery, which needs no remaining substrate in the fermentation broth [115]. Many works on terpenoid biosynthesis by engineered *S. cerevisiae* strains were performed in batch fermentation, mainly in flask assays. For instance, monoterpenes, such as limonene [80,81], geraniol [88], and sabinene [70] were produced in shake flask fermentations by engineered *S. cerevisiae* strains in the range between 0.49 and 17.5 mg/L (Table 3). Moreover, a biosynthetic alternative to D2 diesel (gasoil), sesquiterpene bisabolone, was accumulated by more than 900 mg/L in shake flask fermentation [66]. More recently, other sesquiterpenes, nerolidol [19] and the sedative valerenic acid [82], and polpunonic acid [20], the precursor of the anti-obesity agent diterpene celastrol, have been produced in the range between 1.4 and 336.5 mg/L by highly engineered strains in shake flask fermentations. Concerning batch fermentations in a bioreactor, one example was the production of (S)-linalool, which was accumulated in a tank at 0.26 mg/L [68].

Interestingly, immobilized cell method (ICM) was employed for improving production of some terpenes in flask assays. Immobilized cells prepared by agarose entrapment and free cells (FCM) of *S. cerevisiae* were compared in shake flask fermentations for citronellol production [116]. It was observed that ICM yielded the main product, citronellol (48.5%), whereas FCM accumulated more products along with citronellol, which just yielded 24.2% and 27.2% at incubation time of 6 and 15 days, respectively [116]. Higher citronellol production by ICM was explained by possible improvement of specific metabolic activities [117]. 

Although batch fermentation has several advantages, as explained above, some limitations also exist. The main disadvantage is the growth inhibition of microorganism due to product toxicity, especially in the production of monoterpenes. This problem can only be overcome by continuous removal of the inhibitor product (e.g., limonene) from culture during fermentation, for example, by a two-phase system or by headspace removal [54]. Alternatively, fed-batch fermentation with daily broth draw can be employed.

### 6.2. Fed-Batch Mode

Fed-batch mode is the mostly used operation in the industrial production of diverse type of products as it combines the advantages of both batch and continuous fermentations [103,118]. It has some advantages over batch mode, such as less inhibitory effect of higher accumulated product and feedstock concentration, higher product formation, extended lifespan of cells, maximum cell viability, and easy control of several important physiochemical factors such as temperature, pH, and oxygen saturation [103,119]. Other relevant advantages are the possibility of attaining of high cell density, leading to high product formation, and daily product availability. It is generally considered that fed-batch mode with intermittent removal of product and feeding of feedstock when leftover nutrients are exhausted, is the most effective operation mode for terpenoids since these products become less toxic within this approach. In addition, this process can achieve the highest volumetric productivity. The most critical parameter in the fed-batch mode is the optimization of feeding since it plays an important role for increasing product yield and productivity [83]. 

Many pharmaceutical terpenoids have been produced by fed-batch fermentation (Table 3). The monoterpenes, geraniol [69,73] and limonene [72], produced by engineered *S. cerevisiae* strains, have achieved the titer of 1.69 g/L and 0.918 g/L, respectively, with glucose/ethanol mixture or pure ethanol feeding in fed-batch fermentations. These values were quite low (3 to 6-fold lower) when batch fermentations were employed in the same studies. In these works, two strategies were applied to improve productivity of monoterpenes during fed-batch fermentation. Firstly, a two-phase fermentation system was employed using the nontoxic extractive solvents dodecane and isopropyl myristate to alleviate the toxicity of geraniol and limonene. Then, the dynamic control of ERG20 was attempted by replacing ERG20 promoter with the glucose sensing HXT1 promoter, which is stronger in the presence of glucose and weaker in the absence of glucose. Thus, replacement of HXT1 promoter provided the diauxic fermentation process, where glucose was consumed during cell growth with ethanol production, which was then used for monoterpenes biosynthesis. This approach improved the productivity of geraniol and limonene when glucose/ethanol (1/7), or pure ethanol, were fed instead of glucose as sole carbon source during fed-batch fermentations [69,72], indicating that the dynamic control of ERG20 by HXT1 promoter could provide flux distribution between cell growth and monoterpene synthesis in the absence or presence of low glucose. Moreover, pure ethanol feeding enhanced monoterpene production, since it supplies direct Acetyl-CoA precursors. 

Apart from monoterpenes, great improvement in the production of sesquiterpenes has been achieved using fed-batch processes. For instance, titers of artesiminin precursors amorpha-4,11-diene [60] and artemisinic acid [27] have reached >40 g/L and 25 g/L, respectively, in fed-batch fermentations using highly engineered *S. cerevisiae*. In these studies, the exponential feeding rate for glucose/ethanol mixture substrate feeding and pulse feeding for ethanol substrate (10 g/L) were applied and feeding algorithms were automatically triggered through responses of stir rate, DO percentage as well as evaluation of CO_2_ rate [27,60]. As extracting solvents, isopropyl myristate and methyl oleate were used for artemisinic acid and amorpha-4,11-diene extractions from fermentation broth, respectively [27,60]. Ref. [60] deleted the GAL1, GAL10, and GAL7 gene cluster, as well as GAL80, to prevent galactose use, which is an expensive inducer, in the production of amorpha-4,11-diene. In this attempt, low level of galactose was required to induce production at the end of batch phase of glucose limited fed-batch fermentation. In addition, same authors applied phosphate limited fed-batch fermentation, which could limit growth of cells and channel carbon flux to product formation, ending up with more than 2 times of amorpha-4,11-diene production. Finally, ethanol-restricted fed-batch fermentation was improved, resulting in 37 g/L of amorpha-4,11-diene, which was lower than ethanol unrestricted fed-batch fermentation but this process required lower levels of oxygen uptake rate (OUR), what is desired and feasible for industrial scale fermentation. On the other hand, ref. [27] used isopropyl myristate solvent to extract artemisinic acid from fermentation and improved its productivity in fed-batch fermentations. With this method, a significant enhancement on titer and yield was observed by eliminating successfully the acid from the aqueous phase, since artemisinic acid precipitated in fermentation broth, complicating sampling and negatively affecting viability of cells. Other pharmaceutical sesquiterpenes, such as bisabolene [76], nerolidol [84], patchoulol [22], zerumbone [77], and α-humulene [4] were accumulated by engineered *S. cerevisiae* strains with titers of 5.2 g/L, 5.5 g/L, 0.467 g/L, 0.040 g/L and 1.7 g/L, respectively, in fed-batch fermentations. Constant feeding rate triggered by pH rises was used in bisabolene production of glucose fed-batch fermentation [76] while pulse feeding, which was programmed using script controller feeding and triggered by DO spikes, was performed in nerolidol production [84]. Moreover, carbon-source controlled three-stage fed-batch fermentation was employed for patchoulol production [22], whereas in zerumbone and α-humulene production, constant feed rate triggered by DO and pH rises was applied [4,77]. In these studies, combination of constitutive copper-inducible and diauxic-induced transcription regulation patterns was employed to improve nerolidol production in the fed-batch fermentation through enhancing the growth rates in each phase (exponential, ethanol growth, and diauxic phases) and productivities after the diauxic shift. Thus, combined system yielded better nerolidol titer in the carbon-overflow fed-batches than substrate restricted-fed batches. The improvement in nerolidol productivity might be due to MVA pathway response to “overflow” metabolism, as is seen in ethanol/acetate/glycerol production through glycolysis [84]. Moreover, a carbon source controlled three stages fermentation (first stage: high glucose concentration feeding; second stage: lower concentration of glucose and glycerol feeding; third stage: ethanol feeding) was used to balance the trade-off between the competitive squalene and patchoulol pathways in the production of patchoulol by engineered *S. cerevisiae* [22]. 

The pharmaceutical diterpenes, geranylgeraniol [89], miltiradiene [78], and oxygenated taxanes (precursors of paclitaxel) [90] were also accumulated (3.31 g/L, 0.488 g/L and 0.033 g/L, respectively) by genetically modified *S. cerevisiae* strains (co-culture with *E. coli* in oxygenated taxanes accumulation) in fed-batch fermentations. In these studies, after depletion or decrease of substrate in the medium, feed rates of 5.6 g/h of glucose/ethanol mixture for geranylgeraniol, 5 mL/h of glucose and other ingredients for miltiradiene, and feed pulses of 20 g/L glucose and 50 g/L xylose for oxygenated taxanes productions were applied. 

In addition, several triterpenes and their precursors, such as protopanaxadiol (1.189 g/L) and dammarenediol-II (1.548 g/L) [79], β-amyrin (0.139 g/L and 0.108 g/L) ([71,85], respectively), betulinic acid (0.182 g/L) [75] and ginsenoside Rh2 (2.25 g/L) [35] were attempted to accumulate and successfully produced in fed-batch fermentations. In these studies, distinct feeding strategies were applied. For protopanaxadiol, dammarenediol-II, and ginsenoside Rh2, glucose was fed when ethanol concentration was less than 0.5 g/L, for β-amyrin, glucose at 5 mg/L was fed every 12 h, and for betulinic acid, pulsed glucose and continuous ethanol feeding, controlled by DO signal with ethanol response, was adopted. Highest betulinic acid titer was achieved by excess ethanol feeding, following with nitrogen-limited resting cell fed-batch fermentation [75]. The improvement in this fermentation was possibly due to the increased supply of the precursor acetyl-CoA, or the redox cofactor NADPH, of which 18 and 17 mol were needed for production of 1 mol of betulinic acid [75]. In the same study, solid liquid extraction using polar solvents such as, acetone or ethyle acetate, and subsequent precipitation with strong acid, was applied to recover lupane-type triterpenoids with high selectivity and yield.

It can be deduced from these works that different feeding strategies in fed-batch fermentation depend on the response of engineering strains used and their genetic design, such as the choice of promoters. This is because each strain can have different kind of promoters regulating heterologous gene expression in terpenoids pathway, which may influence the substrate type and the concentrations to be employed in the culture medium. Namely, modifications in inducible gal-promoters and constitutive promoters such as PGPD (glyceraldehyde-3-phosphate), PTEF1 (translational elongation factor), PADH1 (alcohol dehydrogenase), PPGK1 (phosphoglycerate kinase), and uptake control mutations, were reported to affect the feeding strategies in fed-batch terpenoids production [57].

### 6.3. Continuous Mode

Continuous processes (or chemostat cultivations) have been employed in industrial bioprocesses (e.g., insulin production) and have several advantages over batch and fed-batch, such as decreased costs of bioreactor constructions, minimized plant maintenance and operation costs, convenient bioprocess control, requirement of less downtime for tank cleaning and maintenance, and higher productivities [83,103]. One important advantage of this process is to provide a precise comparison of productivities of selected engineered strains under constant conditions and to investigate the effect of the growth rate independently of the other parameters [61]. However, only some studies have applied chemostat cultivation for terpenoid production. Ref. [61] employed continuous cultivation and achieved α-santalene productivity of 0.036 Cmmol/(g biomass)/h at the dilution rate of 0.05 h^−1^. In another study, squalene accumulation of 30 mg/g (product/dry cell weight) was obtained at low dilution rates between 0.05 h^−1^ and 0.20 h^−1^ in glucose limited chemostat cultivation [120]. Despite demonstrated applicability in terpenoids production, continuous fermentation has been poorly explored, mainly because fed-batch fermentation has been shown to be the most effective way for obtaining meaningful terpene titers for industry, combining the advantages of batch and continuous mode at once.

## 7. Conclusions and Future Perspective

Pharmaceutical terpenoid production has attracted increasing attention in recent years because these compounds play significant roles in the prevention and treatment of different types of diseases. As the world’s mortality rate rises, due to the incidence of severe diseases, such as cancer and malaria, the demand for these products increases, and, thus, fast development of alternative and sustainable sources rather than natural and chemical ones is required. Microbial fermentation of these special compounds promises a sustainable, cost effective, and high yield method in contrast to traditional methods. *S. cerevisiae* has been proven to be an efficient cell factory for large-scale terpenoid production. Industrial production of artemisinic acid and β-farnesene are examples of success. 

Process optimization by screening different fermentation strategies is essential to improve TYP of pharmaceutical terpenoids, along with fine-tuning metabolic engineering in *S. cerevisiae*, as this yeast cannot produce target terpenoids naturally. Commercial terpenoid application is only possible with successful combination of these two fields, metabolic engineering and fermentation process development. Process optimization can increase productivity and reduce the cost of production by improving medium composition, physicochemical conditions, and applying efficient downstream processing. Most of the terpenoid titers obtained are in mg/L levels, mainly in batch fermentations, which do not have commercial meaning. On the other hand, bioreactor fermentation with fed-batch operation mode can provide titers at g/L scale, and high productivities, while alleviating the toxicity of final product, which constitutes the most promising approach for industrial production of pharmaceutical terpenoids. Among factors affecting terpenoids fermentation, the pH and DO are relevant factors for certain production cases. Although pH range between 4 and 6 is optimal for yeast growth, and achieve high cell density, some terpenes can be insoluble under this condition, which might complicate the downstream processing. However, this can be solved, for instance, by harvesting tanks at high pH by addition of base solution. The use of extraction agents during cultivation is an option to reduce the toxic effects of terpenes, but it brings extra costs to the process. This problem can be overcome by optimizing the feed strategy together with product removal in fed-batch fermentation. The DO is often used as a trigger in fed-batch process for the control of yeast growth and product formation. Optimum oxygen concentrations are required to balance yeast health and productivity, since low levels can minimize yeast capacity, while high levels can lead to oxidative stress, which can be toxic to cells. Similarly, high temperatures can decrease cell viability, while at low temperatures, high cell density is achieved later. Optimization of media ingredients is also of huge importance as they can account for a significant part of the process costs. One way to decrease feedstock cost could be to find and adapt cheap carbon sources, like agricultural by-products, in *S. cerevisiae* metabolism, as extensively investigated in case of other biomolecules production, which also contributes to process sustainability. Indeed, the trend is for the increase of pharmaceutical terpenoids obtained through optimized fermentation. The power of synthetic biology tools and advances in metabolic engineering strategies for the introducing novel terpenoid biosynthesis pathways in *S. cerevisiae* will bring new opportunities to produce complex terpenes such as the current trend product “cannabinoids”. The microbial fermentation of these highly promising and rare compounds at an affordable cost and high purity, independently from Cannabis plant cultivation, could absolutely support the pharmaceutical market, when the growing industrial demand for these products is foreseen.

## Figures and Tables

**Figure 1 pharmaceuticals-14-00295-f001:**
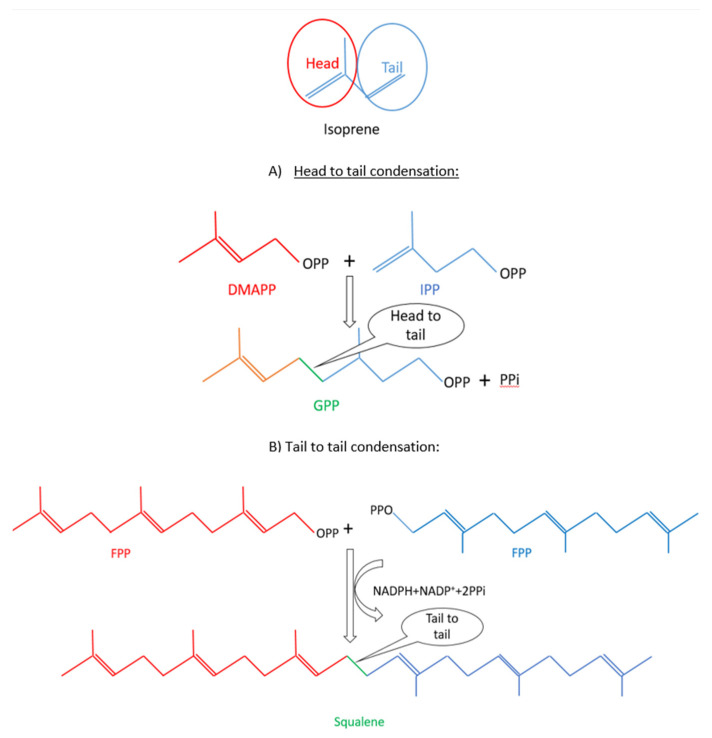
The condensation reactions of terpenoid biosynthesis: (**A**) Head to tail condensation and (**B**) Tail to tail condensation (Adapted from [7]).

**Figure 2 pharmaceuticals-14-00295-f002:**
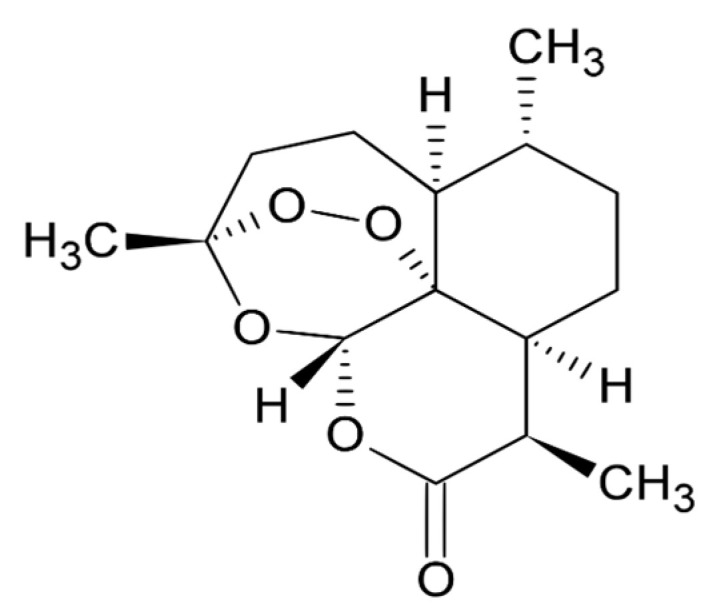
Chemical structure of artemisinin.

**Figure 3 pharmaceuticals-14-00295-f003:**
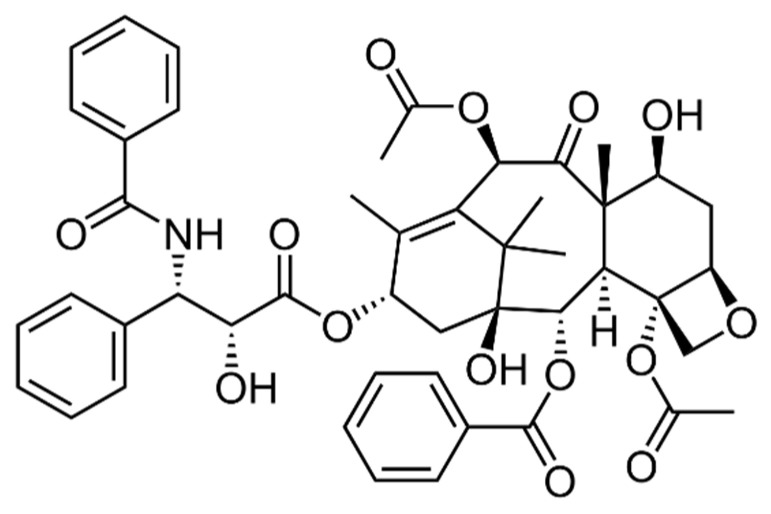
Chemical structure of Paclitaxel.

**Figure 4 pharmaceuticals-14-00295-f004:**
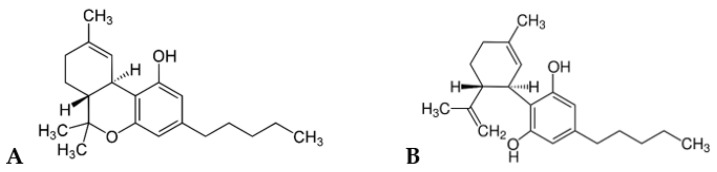
Chemical structure of Δ9-tetrahydrocannabinol (THC) (**A**) and cannabidiol (CBD) (**B**).

**Figure 5 pharmaceuticals-14-00295-f005:**
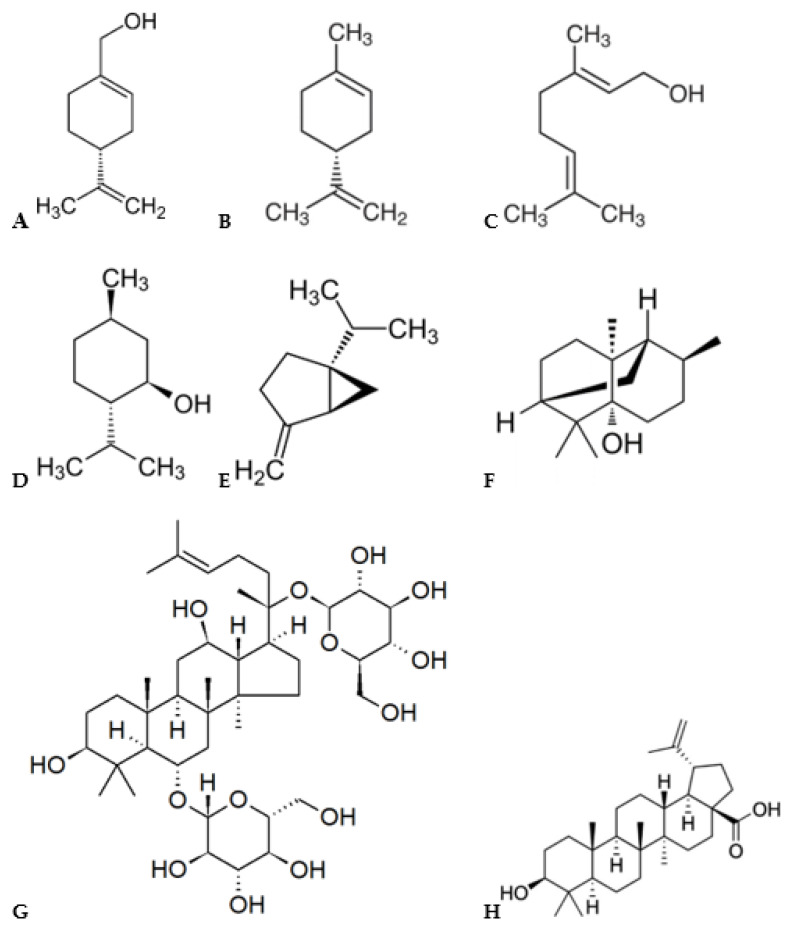
Chemical structure of (**A**) Perillyl alcohol, (**B**) D-limonene, (**C**) Geraniol, (**D**) Menthol, (**E**) Sabinene, (**F**) Patchouli alcohol, (**G**) Ginsenoside and (**H**) Betulinic acid.

**Table 1 pharmaceuticals-14-00295-t001:** Pharmaceutical activities of common terpenoids produced by biotechnological means.

Classification of Terpene	Terpene Name	Pharmaceutical Function	References
Monoterpene	Perillyl alcohol	Anticancer	[3]
	Geraniol	Anticancer	
	D-limonene	Anticancer, transdermal absorption of drugs	
	Menthol	Antimicrobial, transdermal absorption of drugs	
	Sabinene	Antimicrobial	
Sesquiterpene	Artemisinin and its derivatives	Antimalarial, anticancer, antibacterial, antiviral activities and hypoglycemic effect	[3,32]
	Patchoulol	Antibacterial activity	
Diterpene	Paclitaxel	Anti-ovarian, breast, colorectal, head and neck cancers, small-cell and non-small-cell lung cancers (NSCLCs), and treatment of AIDS	[3,8,14,33]
Meroterpene	Cannabinoids	Treatment of pain relieving conditions (in cancer chemotherapy, AIDS, and multiple sclerosis)	[8,34]
Triterpene	Ginsenosides	Anti-oxidation, anti-inflammatory, hepatoprotection, anti-diabetic (hypoglycemic activity) and anti-tumor	[3,35,36]
	Betulinic acid and its derivatives	Anticancer, anti-inflammatory, anti-diabetic, antimicrobial and anti-human immunodeficiency virus (HIV)	[37,38]

**Table 2 pharmaceuticals-14-00295-t002:** Relevant engineering strategies performed in *Saccharomyces cerevisiae* for terpenoid production (adapted from [7]).

Compound	Titer	Strategy	References
Amorpha-4,11-diene	>40 g/L	Overexpression of ADS, upc2-1 Integration of genes copies using control of galactose-inducible promoters: 3 copies of tHMG1, and ERG10, 13, 12, 8, and IDI1 Deletion of gal80Δ Downregulation of ERG9	[60]
Artemisinic acid	25 g/L	Overexpression of ADS, CYP71AV1, CPR1, CYB5, ALDH1 and ADH1 from *A. annua,* HEM1, and CTT1 under control of galactose-inducible promoters Deletion of gal80Δ Downregulation of ERG9	[27]
Farnesene	130 g/L	Overexpression of ADA, xPK, PTA, NADH-HMGr, Farnesene synthase Deletion of adh1 Δ, ald4 Δ, ald6Δ, gpp1Δ, gal2Δ, bdh1Δ Overexpression of enzymes of the MVA pathway to Erg20 Downregulation of ERG9	[26]
Bisabolone	>900 mg/L	Overexpression of tHMG1, ERG20, upc2-1 and bisabolene synthase Downregulation of ERG9	[66]
Alpha-Santalene	92 mg/L	Overexpression of tHMG1 Deletion of Ipp1Δ, dpp1Δ Downregulation of ERG9	[61]
Patchoulol	42.1 mg/L	Overexpression of ERG20 and PatTps177 Downregulation of ERG9	[67]
(S)-Linalool	0.26 mg/L	Overexpression of Erg20 and (S)-linalool synthase Diploid	[68]
Geraniol	1.69 g/L	2μ plasmid of PTEF1-tVoGES-(GGGS)-ERG20WW fusion protein 2μ plasmid of PTEF1-tHMG1, PPGK1-IDI1, PTEF1- upc2.1 PHXT1-ERG20, oye2Δ	[69]
Sabinene	1.75 mg/L	2μ plasmid of PTDH3-ERG20 (F96W-N127W)- sabinene synthase (*Salvia pomifera*) fusion protein Diploid ERG9/erg9, ERG20/erg20, PGal1-HMG2 (K6R), PTDH3-HMG2 (K6R) × 2	[70]

**Table 3 pharmaceuticals-14-00295-t003:** Terpene titers obtained in *S. cerevisiae* and respective fermentation strategies.

Group	Product	Titer (mg/L)	Carbon Source	Nitrogen Source	pH	°C	Aeration /Agitation	Dissolved Oxygen	Inoculum Size	Feeding Strategy	Operation Mode	References
Monoterpenes	Limonene	1.48	2 g/L glucose and 18 g/L galactose	(NH_4_)_2_SO_4_	-	30	200 rpm	-	^1^ OD = 0.05	-	Shake-flask	[80]
	(-)-Limonene	0.49	20 g/L glucose and 20 g/L galactose	(NH_4_)_2_SO_4_	-	30	300 rpm	-	OD = 0.05	-	Shake-flask	[81]
	Limonene	918	20 g/L initial glucose and 10 g/L pure ethanol	Tryptone, Yeast extract	-	30	250 rpm	-	OD = 0.2	Pure ethanol	Fed-Batch in shake flask	[72]
	Geraniol	5	10 g/L glucose	(NH_4_)_2_SO_4_	-	28	-	-	2.5%	-	Shake-flask	[88]
	Geraniol	293	20 g/L initial glucose and then fed solution (glucose and other nutrients)	(NH_4_)_2_SO_4_	6.0	30	600 rpm/1 vvm	>30%	OD = 0.15	Fed solution (glucose and other nutrients) feeding by controlling the specific feed rate to 0.1 h^−1^	Fed-Batch	[69]
	Geraniol	1680	Initial YPD medium, then glucose and ethanol	Yeast extract, Peptone	5.7	30	300–500 rpm /2 vvm	>30%	10%	Glucose feeding under 1 g/L and ethanol feeding under 5 g/L	Fed-Batch (Carbon restricted)	[73]
	Geraniol	1690	Initial 20 g/L glucose then pure ethanol feeding	(NH_4_)_2_SO_4_	5.0	30	600 rpm/1 vvm	>30%	OD = 0.2	400 g/L pure ethanol at 0.1 L/h feed rate	Fed-Batch	[69]
	Sabinene	17.5	20 g/L glucose 20 g/L galactose and 10 g/L raffinose	(NH_4_)_2_SO_4_	-	-	-	-	-	-	-	[70]
	(S)-linalool	0.26	20 g/L glucose	Yeast extract	5.5	30	400 rpm/2 vvm	-	OD = 0.05	-	Batch Bioreactor	[68]
Sesquiterpenes	Bisabolone	900	2 g/L glucose and 18 g/L galactose	Yeast extract, peptone	-	30	180 rpm	-	OD = 0.05	-	Shake-flask	[66]
	Bisabolone	5200	Initial 15 g/L glucose	(NH_4_)_2_SO_4_	5.0	30	0.7 L/min air	-	OD = 0.1	Constant feed rate with pH rise trigger	Fed-batch	[76]
	Nerolidol	336.5	20 g/L sucrose	(NH_4_)_2_SO_4_	-	30	180 rpm	-	OD = 0.2	-	Two-phase flask cultivation	[19]
	Nerolidol	5500	Initial 20 g/L glucose, and then sucrose feeding	(NH_4_)_2_SO_4_	5.0	-	300–600 rpm/1.58–3.16 L/h air	30%	OD = 0.2	Exponential feeding (3 mM sucrose/g biomass/h) with specific increasing rate (0.05 h^−1^ then 20 g/L sucrose pulse feeding by DO spikes	Fed-Batch	[84]
	Valerenic acid	4	20 g/L galactose and 2 g/L dextrose	Yeast extract, Peptone	-	30	200 rpm	-	-	-	Milliliter plates	[82]
	Polpunonic acid	1.4	20 g/L galactose and 10 g/L raffinose	-	-	30	150 rpm	-	-	-	Shake-flask	[20]
	Amorpha-4,11-diene	>4000	Initial 20 g/L glucose then pure ethanol pulse feeding, 0.25 g/L methionine as inducer	(NH_4_)_2_SO_4_	5.0	30	1 L/min air	40%	-	Ethanol pulse feed (10 g/L), Off-gas CO_2_ evaluation rate control	Fed-Batch	[60]
	Artemisinic acid	25,000	Initial 20 g/L glucose then pure ethanol pulse feeding, 0.25 g/L methionine as inducer	(NH_4_)_2_SO_4_	5.0	30	1 L/min air	40%	-	Etanol pulse (10 g/L) feed, stir rate control	Fed-Batch	[27]
	Patchoulol	467	Initial glucose (25 g/L) feeding, Feeds: (1) Sole glucose feeding (2) glucose/glycerol feeding, (3) Sole ethanol feeding	(NH_4_)_2_SO_4_, Peptone. Yeast extract	5.5	30	200–500 rpm/1–2 vvm	-	14%	carbon-source (glucose) controlled three-stage fermentation	Fed-Batch	[22]
	Zerumbone	40	Initial 20 g/L glucose, then feed solution (glucose, other nutrients and ingredients)	Peptone. Yeast extract	5.5	30	300–600 rpm/2 vvm	>30%	5%	Fed rate solution (glucose and other nutrients, ingredients) of 2 mL/min control by DO and pH rise trigger	Fed-Batch	[77]
	α-humulene	1700	Initial 20 g/L glucose, then feed solution (glucose, other nutrients and ingredients)	Peptone. Yeast extract	5.5	30	300–600 rpm/2 vvm	>30%	10%	Fed rate solution (glucose and other nutrients, ingredients) of 2 mL/min control by DO and pH rise trigger	Fed-Batch	[4]
	α-santalene	0.036 Cmmol (g/biomass/h)	Initial 10 g/L glucose and continuous glucose and dodecane feeding	(NH_4_)_2_SO_4_	5.0	30	600 rpm/1 vvm	>30%	^2^ Xi = 1 g/L	Two phase feeding (organic phase and 10 g/L glucose), dilution rates of 0.05/h and 0.1/h	Continuous	[61]
Diterpenes	Geranylgeraniol	3310	Initial 1 g/L glucose, then sole glucose and glucose/ethanol mix. feeding	(NH_4_)_2_SO_4_ corn steep liquor	5.5	33	900 rpm/1 vvm	-	-	Glucose (50% wt/v) and then glucose/ethanol (25% wt/v/50% *v*/*v*) feeding (rate of 5.8 g/h)	Fed-Batch	[89]
	Miltiradiene	488	Initial 20 g/L glucose, then feed solution(glucose, other nutrients and ingredients)	Peptone. Yeast extract	5.5	30	600 rpm/5 L/h air	-	OD = 0.05	Feed solution (glucose and other nutrients) addition by 5 mL/h feed rate.	Fed-Batch	[78]
	Oxygenated taxanes	33	Initial 40 g/L glucose or 20 g/L xylose	(NH_4_)_2_HPO_4_ Yeast extract	7.0	30 and 22	280–800 rpm/0.5 L/min	30%	1% for *E. coli* and 2% for *S. cerevisiae*	Pulse feeding by carbon source control (20 g/L of glucose feed when glucose below 20 g/L and 50 g/L xylose feed when xylose conc. below 10 g/L)	Fed-Batch (co-culture)	[90]
Triterpenes	Protopanaxadiol and dammarenediol-II	1189 and 1548	Initial 25 g/L glucose and then glucose feeding	(NH_4_)_2_SO_4_	5.5	30	1000 rpm/5 L/min air	-	OD = 0.5	Fed solution (glucose and other nutrients) addition when ethanol below 0.5 g/L	Fed-Batch	[79]
	β-amyrin	139	Initial 20 g/L glucose and then pulse ethanol feeding	(NH_4_)_2_SO_4_	-	30	1 vvm	-	OD = 0.2	Pulse ethanol (5 g/L) feeding at every 12 h	Fed-Batch	[71]
	β-amyrin	108	Initial 20 g/L dextrose, and then pulse glucose feeding	Yeast extract Peptone	6.0	30	-	-	-	Pulse glucose (5 mg/L) feeding at every 12 h	Fed-Batch	[85]
	Betulinic acid	182	Initial 50 g/L glucose and then pulse ethanol feeding	NH_4_Cl	6.0	30	1 vvm	>30%	Xi^2^ = 0.08 g/L	Pulse ethanol (25 g/L) feeding control with DO spikes	Fed-Batch	[75]
	Ginsenoside Rh2	2250	Initial 25 g/L glucose and then glucose feeding	Yeast extract Peptone	5.0	30	-	>30%	11%	Fed solution (glucose and other nutrients) addition when ethanol below 0.5 g/L	Fed-Batch	[35]

^1^ Initial optical density (OD); ^2^ Initial biomass (Xi); “-” Not described.

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
