# Peer review of "Fermentation Strategies for Production of Pharmaceutical Terpenoids in Engineered Yeast"

_pharmaceuticals, 2021, doi:10.3390/ph14040295_

Round 1

Reviewer 1 Report

The manuscript submitted for consideration describes fermentation and biotransformation production route of several terpenoids. The review article is comprehensible and condensed. It describes in great detail all aspects of fermentation processes and important methods of preparing biologically active terpenes that are used as drugs. I have just a few small notes and recommendations for the article:

  1. References should be written by authors in a rationalized form and in ascending order. Not a list of e.g. [1] [2] [3], but [1-3]. In the text, it seems a bit rough and confusing. (It appears several times in the manuscript.)
  2. More structural formulas of the mentioned terpenoids would be useful in the text. Some of the most important ones are listed, but others are missing in the text. The structure shown in Fig.3 is not in the same resolution as the others. In contrast, stating the molecular formula of taxol is unnecessary.
  3. line 255: division of word: meth-ylerythritol might be inappropriate, maybe better will be methyl-erythirtol.

Fermentation processes are a very interesting way of preparation. However, for some biologically active substances, it is more advantageous to use extraction methods followed by crystallization to obtain very pure materials. This strategy has proved successful, for example, with betulinic acid, which can also be synthesized by biotransformations. In addition, it may be appropriate to say more about this compound (Betulinic acid), as its semisynthetic derivatives have very interesting effects against HIV and cancer, such as PA-457. In addition, very interesting non-traditional structures can be prepared by biotransformation of BA.

I fully recommend the submitted manuscript to be considered as accept in present form.

Author Response

Point by point response to reviewers - manuscript pharmaceuticals-1157676

The author’s responses are written in blue after the reviewer’s comments and the changes in the revised manuscript are marked in “Track Changes”.

Reviewer #1: The manuscript submitted for consideration describes fermentation and biotransformation production route of several terpenoids. The review article is comprehensible and condensed. It describes in great detail all aspects of fermentation processes and important methods of preparing biologically active terpenes that are used as drugs. I have just a few small notes and recommendations for the article:

  1. References should be written by authors in a rationalized form and in ascending order. Not a list of e.g. [1] [2] [3], but [1-3]. In the text, it seems a bit rough and confusing. (It appears several times in the manuscript.)

Groups of references cited throughout the review were placed in the revised version of the manuscript as indicated by the reviewer.

  1. More structural formulas of the mentioned terpenoids would be useful in the text. Some of the most important ones are listed, but others are missing in the text. The structure shown in Fig.3 is not in the same resolution as the others. In contrast, stating the molecular formula of taxol is unnecessary.

According to the reviewer’s suggestions, the following modifications were made in the revised version of the manuscript: the structural formulas of the mentioned terpenoids perillyl alcohol, D-limonene, geraniol, menthol, sabinene, patchouli alcohol, ginsenoside and betulinic acid were included (Figure 5); the resolution of the structure shown in Fig.3 was improved; and the molecular formula of taxol was removed.

  1. line 255: division of word: meth-ylerythritol might be inappropriate, maybe better will be methyl-erythirtol.

The word “meth-ylerythritol” was replaced by the word “methyl-erythirtol” in the revised version of the manuscript, as corrected by the reviewer (line 291).

  1. Fermentation processes are a very interesting way of preparation. However, for some biologically active substances, it is more advantageous to use extraction methods followed by crystallization to obtain very pure materials. This strategy has proved successful, for example, with betulinic acid, which can also be synthesized by biotransformations. In addition, it may be appropriate to say more about this compound (Betulinic acid), as its semisynthetic derivatives have very interesting effects against HIV and cancer, such as PA-457. In addition, very interesting non-traditional structures can be prepared by biotransformation of BA.

The pharmacological properties of betulinic acid and its semisynthetic derivatives were made clear in the review, as suggested by the reviewer. For that, betulinic acid was included in Table 1 and these sentences were added to the revised version of the manuscript (lines 276-281): In addition, betulinic acid (Figure 5) and its semisynthetic derivatives, such as PA-457, have also shown remarkable pharmacological properties, including inhibitory effects against human immunodeficiency virus (HIV) and cytotoxicity activity on several type of cancer cells (Table 1) [37, 38]. Very interestingly, biotransformation of betulinic acid has been continuously investigated aiming at discovering novel derivatives for pharmacological studies [55]”. Works supporting these sentences were included in the reference list of the revised manuscript:

[37] Amiri et al. 2020 Biotechnology Advances, 38: 107409

[38] An et al. 2020. Applied Microbiology and Biotechnology, 104:3339–3348

[55] Chen et al. 2021 Phytochemistry 182,112608

Reviewer 2 Report

The manuscript consist of an interesting review on the status of arts concerning different fermentative approaches for the heterologous production of isoprenoids of pharmaceutical interest.

The paper deals with a topic of significant interest and it provides a comprehensive report of present knowledge about the different strategies for terpenoids production by using the yeast as cell factory.

Moreover, the Authors efficiently highlights the function of pharmaceutical terpenoids and they supply an updated description of different available approaches for the production of the above compounds by recombinant strain of saccharomyces cerevisiae.

The review is exhaustive and the literature data are given and discussed in a clear way. I think it is a worthy paper with interesting information to report.

However, before publication, the Authors should discuss the pros and cons of using S. cerevisiae as a cell factory for isoprenoid production versus using unconventional yeast or bacteria for the same purpose. The following papers could be cited and discussed:

Wang C. et al. (2018). Microbial platform for terpenoid production: Escherichia coli and yeast. Frontiers in microbiology, 9, 2460.

Lin P. C. & Pakrasi H. B. (2019). Engineering cyanobacteria for production of terpenoids. Planta, 249(1), 145-154.

Zhuang X. et al. (2019). Monoterpene production by the carotenogenic yeast Rhodosporidium toruloides. Microbial cell factories, 18(1), 1-15.

Moreover, the findings reported by Igneas et al (2018, https://doi.org/10.1038/s41589-018-0166-5; 2019; https://doi.org/10.1038/s41467-019-11290-x) should be discussed.

Author Response

Point by point response to reviewers - manuscript pharmaceuticals-1157676

The author’s responses are written in blue after the reviewer’s comments and the changes in the revised manuscript are marked in “Track Changes”.

Reviewer #2: The manuscript consist of an interesting review on the status of arts concerning different fermentative approaches for the heterologous production of isoprenoids of pharmaceutical interest. The paper deals with a topic of significant interest and it provides a comprehensive report of present knowledge about the different strategies for terpenoids production by using the yeast as cell factory. Moreover, the Authors efficiently highlights the function of pharmaceutical terpenoids and they supply an updated description of different available approaches for the production of the above compounds by recombinant strain of saccharomyces cerevisiae. The review is exhaustive and the literature data are given and discussed in a clear way. I think it is a worthy paper with interesting information to report.

  1. However, before publication, the Authors should discuss the pros and cons of using S. cerevisiae as a cell factory for isoprenoid production versus using unconventional yeast or bacteria for the same purpose. The following papers could be cited and discussed: Wang C. et al. (2018). Microbial platform for terpenoid production: Escherichia coli and yeast. Frontiers in microbiology, 9, 2460. Lin P. C. & Pakrasi H. B. (2019). Engineering cyanobacteria for production of terpenoids. Planta, 249(1), 145-154. Zhuang X. et al. (2019). Monoterpene production by the carotenogenic yeast Rhodosporidium toruloides. Microbial cell factories, 18(1), 1-15.

The advantages and disadvantages of S. cerevisiae as a cell factory for isoprenoid production, as compared to other unconventional microbial hosts, were discussed in the review, as suggested by the reviewer. For that, these paragraphs were added to the revised version of the manuscript (lines 116-139):

“Besides S. cerevisiae, other microorganisms have been explored for terpenoids production. Among them, E. coli has the most restrict chassis, since its produces natively limited amounts of terpenoids (e.g., quinones) and therefore, the improvement of MEP pathway by engineering enzymes for IPP and DMAPP synthesis, or the introduction of heterologous MVA pathway, is required [23]. In contrast, S. cerevisiae has endogenous MVA pathway, producing high amounts of ergosterol and native cytochrome P450 enzymes for the modification of terpenoids skeleton. Nonconventional yeast Yarrowia lipolytica has been also considered as a suitable yeast to synthesize terpenoids due to its capacity to produce large amount of acetyl-CoA, the initial substrate of MVA pathway [23]. In addition, carotenogenic yeast Rhodosporidium toruloides can naturally accumulate several carotenoids (C40 terpenoids), indicating that it might have high carbon flux through MVA pathway, ensuring pools of intermediates for producing diverse types of terpenes [24]. This yeast can metabolize efficiently both xylose and glucose, and tolerates high osmotic stress, enabling the use of lignocellulosic hydrolysates as feedstock in contrast to S. cerevisiae [24]. Cyanobacteria have also the potential to produce sustainable terpenoids using light and CO2 instead of sugar feedstocks. However, terpenoids titer and productivity obtained are still below industrial levels and further studies to overcome the barriers for efficient conversion of CO2 to terpenoids are needed [25].   

Overall, S. cerevisiae has as main advantage over E. coli and cyanobacteria hosts its intrinsic MVA pathway, and the disadvantage over Rhodosporidium toruloides host the incapacity of using directly lignocellulosic hydrolysates as feedstock. Nevertheless, S. cerevisiae is quite superior to the other microorganisms in respect to higher process robustness, fermentation capacity, plenty of available genetic tools in pathway engineering and genome editing, and proven capacity to attain industrial levels of relevant terpenoids [23]”. The works indicated by the reviewer, namely, Wang et al. 2018, Lin et al. 2019 and Zhuang et al. 2019, were cited and included in the reference list of the revised version of the manuscript (numbers 23, 24 and 25).

  1. Moreover, the findings reported by Igneas et al (2018, https://doi.org/10.1038/s41589-018-0166-5;2019; https://doi.org/10.1038/s41467-019-11290-x) should be discussed.

The findings reported by Igneas et al. 2018 and 2019 were added to the review, as suggested by the reviewer. For that, this paragraph was added to the revised version of the manuscript (lines 362-368): “Interestingly, to expand the diversity of terpenoid structures, C11 terpene scaffolds were produced in S. cerevisiae by engineering dedicated synthases upon identification of a single residue switch that converts C10 plant monoterpene synthases to C11 specific enzymes [62]. More recently, a synthetic orthogonal monoterpenoid pathway based on an alternative precursor, neryl diphosphate, was established in yeast, in which five engineered enzymes were combined with dynamic regulation of metabolic flux to take advantage of the orthogonal substrate potential and improve monoterpenes production [63]”. Also, the works by Igneas et al. were included in the reference list of the revised version of the manuscript (numbers 62 and 63).